# Controlling the Treatment Time for Ideal Morphology towards Efficient Organic Solar Cells

**DOI:** 10.3390/molecules27175713

**Published:** 2022-09-05

**Authors:** Yiwen Hou, Qiuning Wang, Ciyuan Huang, Tao Yang, Shasha Shi, Shangfei Yao, Donglou Ren, Tao Liu, Guangye Zhang, Bingsuo Zou

**Affiliations:** 1Julong College, Shenzhen Technology University, Shenzhen 518118, China; 2College of New Materials and New Energies, Shenzhen Technology University, Shenzhen 518118, China; 3Guangxi Key Lab. of Processing for Nonferrous Metals and Featured Materials and Key Lab. of New Processing Technology for Nonferrous Metals and Materials, Ministry of Education, School of Resources, Environments and Materials, Guangxi University, Nanning 530004, China

**Keywords:** solvent vapor annealing, power conversion efficiency, organic solar cell, morphology

## Abstract

In this work, we performed a systematic comparison of different duration of solvent vapor annealing (SVA) treatment upon state-of-the-art PM6:SY1 blend film, which is to say for the first time, the insufficient, appropriate, and over-treatment’s effect on the active layer is investigated. The power conversion efficiency (PCE) of corresponding organic solar cell (OSC) devices is up to 17.57% for the optimized system, surpassing the two counterparts. The properly tuned phase separation and formed interpenetrating network plays an important role in achieving high efficiency, which is also well-discussed by the morphological characterizations and understanding of device physics. Specifically, these improvements result in enhanced charge generation, transport, and collection. This work is of importance due to correlating post-treatment delicacy, thin-film morphology, and device performance in a decent way.

## 1. Introduction

Organic solar cell (OSC) technology is one of the most promising photovoltaics (PVs) to be commercialized, which can address the energy consumption issue without causing carbon oxide emission, eventually contributing to carbon neutralization [1,2,3,4,5,6,7,8,9,10]. To date, the urgent task for OSC researchers is to improve the power conversion efficiency (PCE) to a more competitive level, though it has been widely reported with > 18% [11,12,13,14,15,16,17,18,19,20,21,22,23,24,25]; the inputs of chemistry design/synthesis or ternary blend construction are unavoidable, yet the field demands a simple and effective way to improve the device efficiency based on mature material systems.

The eternal topic of improving the PCE involves realizing an ideal active layer morphology that is simultaneously favorable to charge generation, transport, and collection [26,27,28,29,30,31,32,33,34]. To do so, post-treatment on the active layer has been proven effective and promising [35,36,37,38,39]. For instance, Liu et al. reported three types of film morphology that cast layers treated in three different ways: no anneal, thermal anneal, and solvent vapor anneal, where solvent vapor anneal enabled type III morphology realized the highest PCE, due to the best charge generation producing the highest short-circuit current density (*J*_SC_) [40]. Meanwhile, a large number of works have reported that solvent vapor annealing is effective to achieving ideal morphology for different types of blend films [41,42,43,44,45,46,47]. These results all suggest that solvent vapor annealing can induce favorable morphology under proper conditions, which is able to realize high PCE. However, for a high-efficiency system composed by polymer donor small molecular acceptors, the solvent vapor time has been rarely discussed, leaving understanding blank for the field. Motivated by this inconspicuous blind spot, we plan to provide some insights on solvent fumigation duration’s impact to OSC performance.

In this work, we carefully compared the effect of different solvent vapor annealing (SVA) times on active layer morphology and the device performance for a well-developed high-efficiency polymer donor small molecular acceptor composed binary blend, called PM6:SY1 [48]. The results show that with 30 s SVA by chloroform (CF), the active layer can produce the best PCE for OSCs, compared with its 0 s and 50 s counterparts (the operation mode is drawn in Figure 1). The efficiency enhancement comes from concurrently increased *J*_SC_ and fill factor (*FF*), and the kept open-circuit voltage (*V*_OC_). The overly annealed film suffers PCE loss due to the decrease in *J*_SC_ and *FF*. The morphology characterization reveals that the properly fumigated active layer has a better interpenetrating network and slightly reduced phase separation, which facilitates the charge transport. The 50-s SVA-treated film has similar network, but much increased pure domain size; thus, it has overly purified phase distribution and poorer charge generation. The device physics investigation confirms the guess, specifically, better charge generation, transport, and collection. The best efficiency (17.57%) here is at the same level reported recently in some high-level journals using similar method and PM6 as donor ^41^. This work successfully correlates the post-treatment, thin-film morphology, and device efficiency, and is instructive for further realizing PCE progress of OSCs.

## 2. Results and Discussion

The variation in blend film’s ultraviolet–visible (UV–vis) absorption, a basic optical property and reflection of aggregation change, is presented in Figure 2a. Proper treatment duration (30 s) can rationally improve the absorbance of the film, indicating that ideal morphology is achieved. When the time elongates to 50 s, the absorbance generally drops, especially the donor part, implying unfavorable destruction of PM6’s network. As for 80 s, the absorbance of acceptor rises to an extremely high level such that over-phase-purification happens while the PM6’s network is indicated to be seriously harmed.

A series of devices based on traditional structures ITO/PEDOT:PSS-TA/PM6:SY1/PDINN/Ag were fabricated to investigate the photovoltaic parameter change [49,50]. The optimal device performances of each kind of treated film are illustrated as the form of current density vs. voltage (*J*–*V*) characteristics in Figure 2b, and details of photovoltaic parameters are summarized in Table 1. The control device without SVA treatment has a moderate *J*_SC_ of 25.49 mA cm^−2^ and an *FF* of 75.7%. The 30-s chloroform (CF)-fumigated active layer enabled OSC presented a clearly improved *J*_SC_ of 26.14 mA cm^−2^ and *FF* of 77.4%, which is responsible for the optimized PCE of 17.57%. When the SVA treatment time prolongs to 50 s, the device efficiency drops to 16.80%, where the *J*_SC_ and *FF* turn back to a lower level. The 20 independent device data are given in Appendix A for clear information. When the duration prolongs to 80 s, the efficiency drops to an even lower level, where the best PCE is only 15.51% (as shown in Appendix A), due to the drop of *J*_SC_ and *FF*. Furthermore, other two constantly used solvents for SVA are investigated and the results are given in Appendix A for reference. Both tetrahydrofuran (THF) and chlorobenzene (CB) cannot appeal with CF.

Then, we performed the external quantum efficiency (EQE) spectra measurement to check the accuracy of photovoltaic test. As shown in Figure 2c and the integrated value in Table 1, the errors are well-controlled within 3%. The profiles suggests that general charge generation is enhanced for SVA treatment devices, though the overly fumigated one suffers some loss compared with the optimal target. 

The morphology understanding, specifically for the SVA-induced phase separation, is enabled by using the atomic force spectroscopy (AFM) measurement and the grazing incidence small-angle X-ray scattering (GISAXS) test [51,52,53,54,55]. The results are displayed in Figure 3. According to the height images, the surface roughness determined by the root mean square (RMS) values for them are 1.63 nm, 1.15 nm, and 1.23 nm, respectively. This is to say that SVA treatment can reduce the aggregation of materials, which is supposed to be beneficial to charge collection at interface and charge generation. Meanwhile, the phase images show that SVA processing helps the formation of interpenetrating nanostructure, a typical favorable morphology feature for OSCs [56,57,58,59]. These results indicate that SVA treatment bears the hope of boosting *J*_SC_ and *FF*. Then, the difference between the treatment time is compared by the fitted GISAXS results. The pure acceptor phase sizes of them are 31.2 nm, 27.5 nm, and 38.3 nm. The reduced domain size of optimally fumigated film is consistent with better charge generation, and its overlay version shows the largest domain size, which is considered harmful to maintaining *J*_SC_ and *FF*, since the recrystallization process and phase purification overwhelmingly happened [35].

Then, we return our attention to device physics to connect the morphology observation and device performances. The efficiencies of charge generation and collection are evaluated by photocurrent density vs. effective (*J*_ph_ vs. *V*_eff_) relationships, as drawn in Figure 4a. As a result, charge generations for 0 s, 30 s, and 50 s SVA-treated devices are 92.1%, 93.2%, and 91.7%; meanwhile, the collection efficiencies are 84.5%, 87.1%, and 83.5%, respectively. The trap-assisted recombination and bimolecular recombination change of the devices are studied by the *V*_OC_ and *J*_SC_ vs. light intensity relationships, as given in Figure 4b,c. The fitted ideal factors are 1.18, 1.05, and 1.22, respectively. Additionally, the bimolecular recombination rates are found to be 0.97, 0.98, and 0.96. These results show that, when pretreated by SVA properly, the recombination can be well-suppressed, which is beneficial to *J*_SC_ and *FF*.

Furthermore, the charge transport improvement is expected to be confirmed, by using the space-charge limited current (SCLC) method. The results of hole-only and electron-only devices are shown in Figure 4d,e, where the linear relationship of applied voltage and squared current density is determined. The calculated hole and electron mobilities (*μ*_h_, *μ*_e_), together with the *μ*_h_/*μ*_e_ ratios, are presented in Figure 2f. It is found that optimal film has the most efficient and balanced charge transport, which facilitates the pursuit of *FF*.

## 3. Materials and Methods

### 3.1. Characterization

AFM measurements were obtained by using a Dimension Icon AFM (Bruker, Billerica, MA, USA) in tapping mode. The grazing incidence X-ray scattering (GIWAXS) measurement was carried out with a Xeuss 2.0 SAXS/WAXS laboratory beamline using a Cu X-ray source (8.05 keV, 1.54 Å) and a Pilatus3R 300 K detector. The incident angle was 0.2°. The samples for GISAXS measurements were fabricated on silicon substrates using the same recipe for the devices.

### 3.2. Solar Cell Fabrication and Characterization

Solar cells were fabricated in a conventional device configuration of ITO/PEDOT:PSS-TA/active layers/PDINN/Ag. The ITO substrates were first scrubbed by detergent; then sonicated with deionized water, acetone, and isopropanol; and dried overnight in an oven. The glass substrates were treated by UV–Ozone for 30 min before use. PEDOT:PSS (Heraeus Clevios P VP AI 4083) doped by pyramine hydrochloride was spin-cast onto the ITO substrates at 4000 rpm for 30 s, and then dried at 150 °C for 15 min in air. The PM6:SY1 (1:1.2) was dissolved in 15 mg mL^−1^ chloroform with 1-chloronaphthalene (0.5% vol) as additive and stirred overnight in a nitrogen-filled glove box. The solution was spin-cast at 3000 rpm for 30 s onto PEDOT:PSS-TA films followed by solvent vapor (saturated CF) annealing for different time. A thin PDINN layer was coated on the active layer, followed by the deposition of Ag (evaporated under 5×10^−4^ Pa through a shadow mask). The optimal active layer thickness measured by a Bruker Dektak XT stylus profilometer (Bruker, Billerica, Massachusetts, United States) was about 100 nm. The current density–voltage (*J*–*V*) curves of all encapsulated devices (by Epoxy) were measured using a Keithley 2400 Source Meter in air under AM 1.5 G (100 mW cm^−2^) using a Newport solar simulator, where the applied voltage varies from −1.3 V to 0.5 V. The light intensity was calibrated using a standard Si diode (with KG5 filter, purchased from PV Measurement to bring spectral mismatch to unity). The scan speed is about 0.5 V/s, and devices were scanned several times to exlude the effect of burn-in loss. Optical microscope (Olympus BX51, Shinjuku City, Tokyo) was used to define the device area (4.0 mm^2^). EQEs were measured using an Enlitech QE-S EQE system equipped with a standard Si diode. Monochromatic light was generated from a Newport 300 W lamp source.

### 3.3. SCLC Measurements

The electron and hole mobility were measured using the method of space-charge limited current (SCLC) for electron-only devices with the structure of ITO/ZnO/active layer/PDINN/Ag and ITO/PEDOT:PSS-TA/active layers/MoO_x_/Ag for hole-only devices. The charge carrier mobility was determined by fitting the dark current to the model of a single-carrier SCLC according to the equation *J* = 9*ε*_0_*ε*_r_*μV*^2^/8*d*^3^, where *J* is the current density, *d* is the film thickness of the active layer, *μ* is the charge carrier mobility, *ε*_r_ is the relative dielectric constant of the transport medium, and *ε*_0_ is the permittivity of free space. *V* = *V*_app_ − *V*_bi_, where *V*_app_ is the applied voltage and *V*_bi_ is the offset voltage. The carrier mobility can be calculated from the slope of the *J*^1/2^~*V* curves.

### 3.4. The Analysis of J_ph_ vs. V_eff_ Relationships

The definition of *J*_ph_ is the current density under illumination (*J*_L_) minus the dark current density (*J*_D_), and *V*_0_ refers to the voltage value when *J*_ph_ = 0. Accordingly, *V*_eff_ = *V*_0_ - *V*_appl_, where *V*_appl_ represents applied voltage, has a clear meaning. Importantly, when *V*_eff_ reaches a high value (> 2V), it is normally believed that generated excitons are fully collected, in which *J*_ph_ is equal to saturated current density (*J*_sat_). Then, we can calculate *J*_SC_/*J*_sat_ and *J*_max_/*J*_sat_ to describe exciton dissociation (*η*_diss_) and charge collection (*η*_coll_) efficiency. *J*_max_ is the *J*_ph_ at the maximal output point.

## 4. Conclusions

In summary, the SVA treatment time is found to be impactful in determining the final morphology feature and device performances. Previous work demonstrated the effectiveness of SVA, but rarely paid attention to the processing time. Thereby, the work herein is meaningful for the field to understand how to realize high PCE. The best device efficiency, as high as 17.57%, was enabled by improved *J*_SC_ and *FF*, and the other two counterparts suffered poorer performance due to unfavorable morphology features, where initially large phase separation was found to be the reason. The device physics investigation confirmed the guess, specifically, better charge generation, transport, and collection. This work is of importance due to correlating the post-treatment delicacy, thin-film morphology, and device performance in a decent way. Possible future works could lie in finding out more proper treating solvents and more suitable photovoltaic systems to achieve higher PCE and even break through the bottleneck of the whole field.

## Figures and Tables

**Figure 1 molecules-27-05713-f001:**
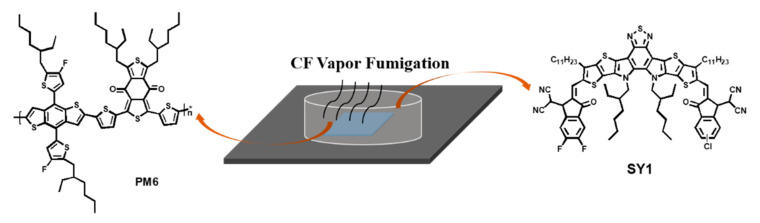
Schematic diagram of CF SVA-treated film based on PM6 and SY1.

**Figure 2 molecules-27-05713-f002:**
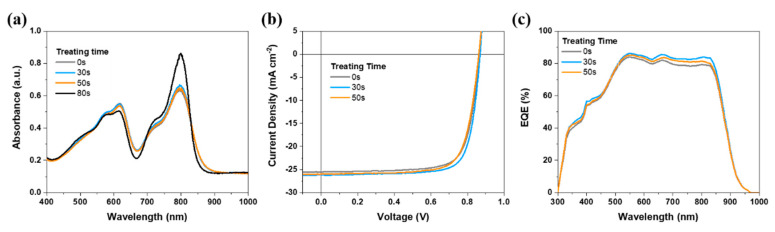
(**a**) UV–vis absorption of active layers. (**b**) *J*–*V* characteristics. (**c**) EQE spectra.

**Figure 3 molecules-27-05713-f003:**
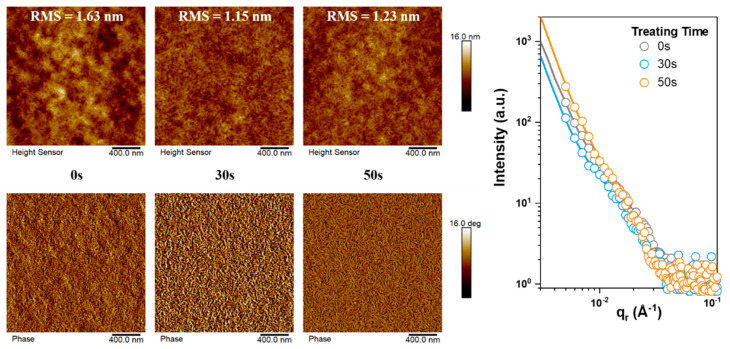
AFM height and phase images, and GISAXS results.

**Figure 4 molecules-27-05713-f004:**
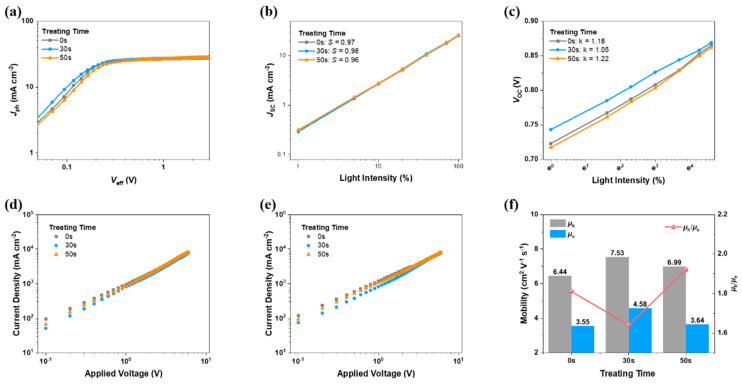
(**a**) *J*_ph_ vs. *V*_eff_ relationships. (**b**) *J*_SC_ and (**c**) *V*_OC_ vs. light intensity plots. Hole-only (**d**) and electron-only (**e**) device results. (**f**) summarized *μ*_h_, *μ*_e_ and ratios.

**Table 1 molecules-27-05713-t001:** Device performances.

PM6:SY1	*V*_OC_ (V)	*J*_SC_ (mA cm^−2^)	*FF* (%)	PCE (%)
SVA 0 s	0.865	25.49/25.89	75.7	16.69 (16.44 ± 0.17)
SVA 30 s	0.869	26.14/25.93	77.4	17.57 (17.30 ± 0.20)
SVA 50 s	0.862	25.97/25.40	75.0	16.80 (16.60 ± 0.10)
SVA 80 s	0.857	25.06/24.63	72.2	15.51

Integrated current density values are behind the slashes. The brackets contain averages and standard errors of PCEs based on at least 20 devices.

## Data Availability

Data are available from authors based on reasonable requirement from readers.

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
