# Peer review of "Controlling the Treatment Time for Ideal Morphology towards Efficient Organic Solar Cells"

_molecules, 2022, doi:10.3390/molecules27175713_

Round 1
Reviewer 1 Report (Previous Reviewer 4)
In the resubmitted manuscript, the authors have improved and have addressed already all concerns. Therefore, I recommend the manuscript for possible publication in the molecules.
Author Response
Thanks for the reviewer's positive comment.
Reviewer 2 Report (Previous Reviewer 1)
I appreciate the authors for their responsible and detailed approach to the analysis of the previous reviewers' comments. In my opinion, the revised manuscript looks more complete, evidence-based and scientifically substantiated.
Perhaps the duplication of information on the solar cells characterization and characterization in the "Materials and methods" part of the manuscript and in the SI is redundant. Instead of repeating this information in the SI, it is worth bringing the device on which the UV-Vis spectra were obtained.
Author Response
We appreciate the reviewer's valuable suggestions, now the revision has been done accordingly.
This manuscript is a resubmission of an earlier submission. The following is a list of the peer review reports and author responses from that submission.
Round 1
Reviewer 1 Report
The manuscript is devoted to the study of the influence of duration of solvent vapor annealing (SVA) treatment to the thin-film morphology and overall photovoltaic characteristics of organic solar cells. The creation of highly efficient organic solar cells not only depends on the chemical structure of their constituent layers, but is also a kind of art, because too many factors affect the final characteristics of such devices. Creating the ideal morphology of certain layers of solar cells by controlling the treatment time is one way to increase their photovoltaic efficiency.
The authors studied the effect of SVA duration on the properties of solar cells based on a donor-acceptor system PM6:SY1. At the same time, the authors studied only two exposure times: 30 s and 50 s. The authors chose only chloroform as the solvent. The authors should explain on the basis of which such an exposure time and this particular solvent were chosen. Also, for a deeper understanding of the ongoing processes, the authors should analyze a larger number of solvents and expand the exposure time range.
In addition, neither the text of the manuscript nor the SI contain a detailed description of the process of constructing organic solar cells, as well as the SVA process, which was studied in detail by the authors. The authors mention that PCE values were obtained as an average of at least 20 devices. Perhaps, for additional confirmation of the convergence of the results obtained, all statistical data should be included in the SI.
The authors used a wide range of modern methods for analyzing and characterizing the properties of both the solar cells themselves and the surfaces of individual layers. To more fully reflect the effect of SVA on the layer morphology, one can also analyze the UV-visibility spectra of thin films obtained with different SVA times.
Among the technical inaccuracies, I would like to note the need to decipher the abbreviation CF at the first mention (p. 2, line 55), as well as the discrepancy between the SVA time given in Table 1 and the SVA time discussed in the text of the manuscript and shown in the figures.
Reviewer 2 Report
Overall, this is an interesting work. Authors should carefully consider the following comments: 1. Fig. 3-4 are too small for readers. It is recommended that these figures should be enlarged to improve the readability. 2-Future studies should be provided.
Reviewer 3 Report
This work reports various interesting results. However, several deficiencies are spotted.
There are a few remarks that, I hope, can help the authors to improve the text and also useful to the readers.
1. The abstract is written in general and may be rewritten.
2. From my view, the introduction is very important in the paper. In the introduction, you should tell the issue and novelty of this paper. However, the motivation part are a bit weak and it is highly recommended to make it more exciting. Please carefully rewrite these two parts.
3. English is needed to be improved a bit more in order to give a clear picture of your achievements.
4. It is better to compare the results with similar cases
5. The result section is good and I understand it exactly.
Reviewer 4 Report
In this manuscript, the authors performed a comparison of different duration of solvent vapor annealing (SVA) treatment upon PM6:SY1 blend film. I am feeling sorry to reject the manuscript. In the abstract authors state "state-of-the-art PM6:SY1 blend film", but the information provided in the manuscript did not support this statement. SVA is a very mature field, it is not new. The authors failed to exploit the novelty in the manuscript. Apparently, the study is looking incomplete.